# Dietary Practices and Adolescent Obesity in Secondary School Learners at Disadvantaged Schools in South Africa: Urban–Rural and Gender Differences

**DOI:** 10.3390/ijerph17165864

**Published:** 2020-08-13

**Authors:** Alice P. Okeyo, Eunice Seekoe, Anniza de Villiers, Mieke Faber, Johanna H. Nel, Nelia P. Steyn

**Affiliations:** 1Department of Nursing Science, University of Fort Hare, Ring Road, Alice 5700, South Africa; aokeyo@ufh.ac.za; 2Sefako Makgatho Health Science University, Ga-Rankuwa 0208, South Africa; Eunice.Seekoe@smu.ac.za; 3Non-Communicable Diseases Research Unit, South African Medical Research Council, Cape Town 7505, South Africa; Anniza.deVilliers@mrc.ac.za (A.d.V.); mieke.faber@mrc.ac.za (M.F.); 4Centre of Excellence for Nutrition, North-West University, Private Bag X1290, Potchefstroom 2520, South Africa; 5Department of Logistics, Stellenbosch University, Stellenbosch 7600, South Africa; jhnel@sun.ac.za; 6Division Human Nutrition, Department of Human Biology, University of Cape Town, UCT Medical Campus, Anzio Road, Anatomy Building, Observatory, Cape Town 7925, South Africa

**Keywords:** adolescents, black African, South Africa, eating practices, BMI, meal pattern

## Abstract

South Africa has a high prevalence of obesity in black female adolescents and a paucity of knowledge regarding contributing dietary practices. The aim of this study was to assess the dietary practices and weight status of male and female adolescents at secondary schools in the Eastern Cape province in urban and rural areas. Sixteen schools and grade 8–12 learners (N = 1360) were randomly selected from three health districts comprising poor disadvantaged communities. A short unquantified food frequency questionnaire was used to collect data on learners’ usual eating practices with regards to weekly meal pattern, breakfast consumption, foods taken to school, takeaways, and snacks eaten while watching television (TV). Body mass index measurements were determined for each learner. Prevalence of combined overweight and obesity differed significantly between genders, 9.9% in males versus 36.1% in females (*p* < 0.001). Significant gender differences were noted regarding eating practices. Females had a higher frequency of eating sugary snacks (*p* < 0.001) and a lower frequency of eating breakfast (*p* < 0.01) than males. Females ate significantly more fried fish (*p* < 0.05), pizza (*p* < 0.05) fat cakes (fried dough balls) (*p* < 0.05), hotdogs (*p* < 0.01), candy (*p* < 0.001), cake (*p* < 0.01), and crisps (*p* < 0.001). Compared to urban areas, the frequency of eating breakfast (*p* < 0.01) and sugary snacks (*p* < 0.05) was significantly higher in rural areas. Significantly more learners in urban areas consumed boerewors (beef sausage) rolls (*p* = 0.027), hamburgers (*p* = 0.004), and soft drinks (*p* = 0.019), while more learners in the rural areas consumed cordial (*p* = 0.001). In conclusion, a high prevalence of combined overweight and obesity was found in black female adolescents and a high prevalence of poor dietary practices was observed, with significant gender and urban–rural differences.

## 1. Introduction

Nittari et al. 2019 [1] have postulated that childhood obesity is the most serious public health problem of this century. They indicated that in the European region one out of three children is overweight or obese. In the USA, currently about 17% of children are obese [2]. In sub-Saharan Africa (SSA) the prevalence of combined overweight and obesity ranges from 2% to as high as 54% across 20 countries and is increasing at an alarming rate [3].

The combined prevalence of overweight and obesity of South African adolescents increased from 29.3% to 36.5% in females and from 7.9% to 14.3% in males between 2002 and 2008, respectively [4]. In both years, the overall prevalence was substantially higher in females than males. In the South African Health and Nutrition Examination Study (SANHANES) of 2012 [5], the combined prevalence of overweight and obesity was 26.8% in 15 to 17-year-old females and 48.8% in 18 to 24 year-old females, implying an even greater increase. A recent review of cross-sectional studies undertaken on South African adolescents in urban and rural settings showed an overall increase in the combined prevalence of overweight and obesity with age in adolescents from age 10 to 20-years, mostly because of an increase in obesity in females [6].

Briefel and Johnson (2004), evaluating secular trends in dietary intake in the United States over 30 years, found that in children 1 to 19-years-old energy intake had increased little except in adolescent females [7]. This was attributed to an increase in the population eating away from home, increased consumption of sugar sweetened beverages (SSBs), larger portion sizes and changes in snacking habits.

Adolescents experience more time away from the family than younger children and hence are exposed to many foods which may not be the norm at home. Growing levels of independence make adolescents more susceptible to the development of obesogenic behaviors which increases the risk of obesity [6]. Numerous studies have examined sedentary behaviors in adolescents, especially television (TV) watching and/or excessive screen time, and found them to be associated with a less healthful diet [8,9,10,11,12].

A recent review indicated that socio-economic status significantly influences eating behaviors, such as excessive snacking, which may lead to excess calorie consumption, in turn potentially leading to obesity [13]. Furthermore, adolescents from disadvantaged groups tend to have the highest rates of obesity, particularly affecting ages 12 to 19-years and may have the chronic cardiometabolic symptoms usually observed in adults, such as hyperglycemia, hypertension, dyslipidemia and inflammation [14]. While cardiovascular risk factors have been extensively studied in adults, there are less data on adolescents. Consumption of mono- and poly-unsaturated fatty acids, dairy products, fiber, vitamin D and fruit and vegetables have shown a positive association with cardiovascular health, while an adverse association has been found with increased intake of saturated fat, sodium, SSBs and fast food [15]. Hill et al. [16] examined the effects of stress on eating behaviors and found that stress was associated with eating behaviors of both younger and older children. They suggested that the effects of stress on unhealthy eating may begin as early as eight or nine years of age.

Kuzbicka and Rachon [17] postulated that poor eating habits, excessive calorie intake, coupled with a sedentary lifestyle are the main causes of obesity in children. They give examples of undesirable eating habits such as eating of highly processed and calorie-rich foods between meals, eating in front of the TV, drinking SSBs, skipping breakfast, eating out frequently, and emotional eating. A few studies have mentioned an increased consumption of sweet and salty snacks, SSBs and/or decreased consumption of fruit and vegetables in the diet as being associated with increased obesity in children and/or adolescents [18,19,20].

The aim of this study was to assess the dietary practices and weight status of male and female adolescents at secondary schools in urban and rural areas of the Eastern Cape Province.

## 2. Materials and Methods

### 2.1. Setting and Sample

The study was conducted in three districts of the Eastern Cape Province (Figure 1), which is the second largest province in the country, with a total population of seven million people in 2017 [21]. The province has 8 districts (2 metro district municipalities and 6 district municipalities) and 23 school districts. There are 5589 public schools in the province [21]. The study population consisted of learners attending public secondary schools in quintiles 1, 2, and 3. Schools in these quintiles, represent the poorest and most disadvantaged schools with lowest socioeconomic status. All the school funds come from the government. Learners in all schools in these quintiles receive a government provided meal daily as part of the National School Nutrition Program.

The sample size for this study was drawn by selecting the number of secondary school learners registered in quintiles 1, 2 and 3 secondary schools. A probability multistage-cluster sampling method was used. This procedure ensured adequate representativeness of the study population in the sample. The procedure involved arrangement of the study population into provincial districts, school districts, schools and grade-level clusters. In the first stage, three provincial districts (1 metro district municipality and 2 district municipalities) were conveniently selected from the 8 provincial districts. These districts have urban and rural schools in quintiles 1, 2 and 3. The second stage involved random selection of two education districts each from the three provincial districts. The third stage involved randomly selecting 18 eligible secondary schools from quintiles 1, 2 and 3 in each education district. It was calculated that this number of schools would provide a representative sample. The fourth stage involved purposive selection of grade level 8 to 12 in each of the selected schools. These grade levels were selected because the target population was secondary school learners. In each of the selected grades, simple random balloting was used to select learner participants. Learners were asked to pick a “Yes” and a “No” paper in randomized manner. Those who had picked “Yes” were given a consent form to be signed by a parent/guardian. Two out of the 18 sample schools were eliminated from the study because learners did not return the consent forms. A total of 1500 learners were selected, but only 1360 participants (528 boys and 832 girls) eventually completed the questionnaire, and their data were used in the final statistical analysis. Data from 143 participants were incomplete and were not included in the analysis. More male learners than female learners did not return their consent forms and were excluded from the study; resulting in an unequal distribution.

### 2.2. Selection and Training of Research Assistants

The study utilized five research assistants who received training during a workshop arranged for this purpose. The research assistants were final year Bachelor of Human Movement Science students in the Department of Human Movement Science, University of Fort Hare, as well as final year students of Food Technology at Walter Sisulu University.

### 2.3. Measurements

#### 2.3.1. Demographic Data on Learners

Information on the learner’s age, gender, ethnicity and grade, geographic location, school quintile and their mother and father’s highest level of education was collected by a questionnaire completed by the learners.

#### 2.3.2. Food Frequency Questionnaire

Dietary practices are defined as: ‘observable actions or behaviors of dietary habit and can be classified as good dietary practices and poor dietary practices’ [22].A shortened unquantified food frequency questionnaire (FFQ) was used to collect data on learners’ usual dietary practices with regards to foods consumed (i) for breakfast, (ii) as takeaways, (iii) and as snacks while watching television (TV). The FFQ used for this study was adapted from Audain et al. [23] developed for adolescents in KwaZulu-Natal, South Africa. The original questionnaire was developed by six dietitians and provided options for 61 foods according to the following groups: starches, vegetables, fruit, dairy, meat, fast food, takeaways, snacks and drinks. Foods included in the current FFQ were those eaten for breakfast, eaten in the school environment, takeaways, fast foods, snacks and drinks. The food list for breakfast had 10 different food options: cooked porridge, instant cereal, milk on porridge/cereal, white bread, brown bread, spread on bread, milk or yoghurt to drink at breakfast, tea/coffee, milk in tea/coffee, fruit. Learners were asked to indicate how frequently they ate the food items during the week and over the weekends respectively. The items selected were based on foods typically eaten for breakfast in the Eastern Cape.

The food list for takeaways had 20 food items: cordial ( sugary beverage made from a concentrate), soft drinks (SSBs), ice-cream, crisps, chocolates, cake/biscuits, sweets (candy), doughnuts, boerewors (local beef sausage) rolls, hotdogs, pita bread, samosas (triangles of pastry filled with curry meat or vegetables), pies, sausage rolls, fat cakes (fried balls of wheat flour dough), pizza, fries, fried fish, chicken burgers, hamburgers. Learners were asked to indicate the take-aways eaten during the past week, while the list for snacks eaten while watching TV had 11 food items (fruit, popcorn, chocolates, bread, crisps, biscuits, cakes, doughnuts, eclairs, soft drinks, fries). Consumption frequency was categorized as: none per week; once to twice times a week; three to four times a week; and five or more times a week. For some of the data analyses, frequency categories were combined. Learners were also asked about their main sources of nutrition information and whether they took a lunch box to school.

#### 2.3.3. Anthropometry

Anthropometric measurements were taken with the learners in light clothing and no shoes, and in accordance with the International Society for the Advancement of Kinanthropometry (ISAK) recommendations [24]. Height was measured to the nearest 0.1 cm using a calibrated vertical stadiometer (Seca PorTable 217 Seca, UK). Weight was measured to the nearest 0.01 kg using a calibrated digital electronic weighing scale (Seca 813, Seca, UK) which was calibrated after every 20th learner. Height and weight measurements were taken in duplicate and the average of the two measurements was calculated.

### 2.4. Pilot Study

A pilot study was completed including 25 learners from two secondary schools (one quintile 1 and one quintile 3 school, respectively) in Buffalo City Metropolitan Municipality. These learners and their schools were not included in the main study. The aim of the pilot study was to ensure the feasibility of the questionnaire and study procedures, and to ensure that question formats and sequences were appropriate for the learners’ cognitive and reading ability. Adjustments were made, where needed.

### 2.5. Data Analyses

The final questionnaires and measurements of 1360 participants data were analysed. Weight and height measurements were used to calculate the body mass index (BMI) (weight(kg)/height(m)^2^), which was used to classify learners as either underweight, normal weight, overweight or obese based on internationally accepted cut-off values [25]. For learners younger than 18 years, age and sex specific cut-off values were calculated using the extended international (IOTF) BMI cut-off values [25], while equivalent adult cut-off values were used for learners 18 years or older [26].

Data on the food items consumed, sources of nutrition information and anthropometric status were analyzed using descriptive statistics and are summarized as frequencies and percentages. Relationships were determined using the Rao–Scott chi-square test and the independent t-test was used to test for differences between two independent groups taking the complex sampling design into consideration; a *p*-value of < 0.05 was considered statistically significant. The Bonferroni test was used to test for differences between age groups. Bivariate logistic regression analysis was conducted to establish relationships between the risk of underweight, overweight/obesity or obesity between sociodemographic characteristics and eating practices, respectively. All statistical analyses were performed with the Statistical Package for Social Science (SPSS) version 21.0 for windows (SPSS) and SAS.

### 2.6. Ethics

Ethical approval for the study was granted by the Research Ethics Committee of the University of Fort Hare (Reference number: JIN011SOKE01). Permission was requested from school principals to collect data from the learners and teachers within the school setting. Dates and time for data collection were arranged through consultations with the schools. Permission to carry out the study was granted by the Departments of Basic Education and Department of Health, Eastern Cape Province, South Africa. Written consent was obtained from learners older than 18 years old, from parents/guardians of younger learners, and from Life Orientation (LO) teachers. In addition, written assent was obtained from all learners younger than 18 years before data collection.

## 3. Results

### 3.1. Demographic Data

Table 1 provides data on the sample group (N = 1360). The age of the learners ranged from 11 to 26 years. The World Health Organization (WHO) defines adolescents as those between 10 and 19 years of age [27]. ‘Other overlapping terms used in this report are youth (defined by the United Nations as 15–24 years) and young people (10–24 years), a term used by WHO and others to combine adolescents and youth’ [27]. For the purpose of this study, we will refer to adolescents since they made up a major part of the sample. Most (96.8%) of the learners were Black African, 61.1% were females and 38.9% males. Fifty-two percent of learners came from rural areas and 48% from urban areas. The numbers of learners were evenly distributed between the grades. More than 20% of mothers and fathers only had a primary education or none.

### 3.2. Anthropometry

Table 2 presents data on the anthropometric data of the learners. Mean BMI is significantly lower in males than in females (20.7 kgm^2^ vs. 23.9 kgm^2^; *p* < 0.001). The prevalence of underweight is higher in males than in females (13.5% versus 5.5%). There is a larger percent of males in the normal weight group compared with females (76.8 vs. 58.4%). For males, 9.9% were either overweight or obese, with no clear trend across the age groups. In contrast, 36.1% of females were either overweight or obese, with the prevalence for both overweight and obesity respectively increasing over the age groups. In females, there is an increase observed from the youngest to the oldest age groups for the prevalence of overweight (19.7% to 25.7%), obesity (7.9% to 14.3%) and combined overweight and obesity (27.6% to 41%). For both males and females, prevalence of combined overweight and obesity was higher in urban than rural learners (males 14.0% vs. 6.3%, *p* < 0.001; females 41.3% vs. 32.4%, *p* = 0.023).

### 3.3. Usual Weekly Eating Pattern Described by the Learners

Table 3 indicates that 67.4% of learners were regular consumers of breakfast (3–5 weekdays) and males ate breakfast more frequently than females did (*p* < 0.01). It is also noted that 72.3% of rural learners were regular breakfast consumers compared with 62.1% of urban ones (*p* < 0.001). This was also the case on weekends (78.2% vs. 66.2%, *p* < 0.001). Around 60% of males and females were regular consumers of three meals a day. Nearly 30% of learners ate three or more snacks a day with more females than males eating snacks (33.5% vs. 24.2%, *p* < 0.01). More urban learners appeared to drink hot drinks three or more times a day with at least two teaspoons of sugar than rural ones. Table 3 also shows that there were numerous significant gender differences in snacking while watching TV, and frequency of eating breakfast. Females had a significantly higher frequency of eating sugary snacks (7.6 times vs. 6.5 times in past week, *p* < 0.01) and salty snacks (1.7 times vs. 1.5 times than males (*p* < 0.01). Males had significantly more healthy (2.5 times vs. 2.2, *p* < 0.05) and unhealthy snacks (9.1 vs. 8.1 times, *p* < 0.05) during the past week while watching TV, and ate breakfast more frequently (3.2 times vs. 3.1, *p* < 0.01) than females. Females took a lunch box to school more frequently than males (2.3 times vs. 1.7, *p* < 0.001). There were few significant urban–rural differences. The frequency of eating breakfast was significantly higher in rural than urban areas (3.2 times vs. 3.0 times, *p* < 0.001).

Table 4 provides data on usual breakfast consumption on weekdays. Learners indicated having cooked porridge with milk (25.2%) and without milk (16.3%), and instant cereal with milk (29%) regularly (3–5 times on weekdays). White bread with spread was consumed by 29.1% and without spread by 16.9% regularly; brown bread with spread by 32.6% and 18.7% without spread. Tea or coffee with milk was drunk by 38.3% on three to five weekdays and without milk by 24.6%. More than 40% of learners had fruit regularly with breakfast. A comparison of gender indicated that the frequency of eating cooked porridge without milk was significantly less in females (*p* < 0.001. There was only one urban–rural difference with less urban children having milk in their tea or coffee than rural ones (*p* < 0.01).

### 3.4. Frequency of Takeaway Food Consumed by Learners in the Past Week

Sweetened beverages (soft drinks and cordial) were most commonly consumed five or more times a week (34%), followed by crisps (32%), sweets (candy) (30%), fat cakes (balls of fried bread dough) (15%) and chocolates (15%) (Figure 2). Only 22% learners did not purchase soft drinks in the past week. Consumption of cooked takeaway food items were generally lower than those of ready-to-eat items. Females ate fried fish (*p* < 0.05), pizza (*p* < 0.05) fat cakes (*p* < 0.05), hotdogs (*p* < 0.01), sweets (*p* < 0.001), cake (*p* < 0.01), and crisps (*p* < 0.001) as takeaways more frequently than males.

Learners in urban areas consumed boerewors (beef sausage) rolls (*p* = 0.027), hamburgers (*p* = 0.004), and soft drinks (*p* = 0.019) significantly more frequently as takeaways, while learners in the rural areas consumed cordial (*p* = 0.001) more frequently; it is a cheaper option than soft drinks.

### 3.5. Snacking Habits While Watching Television

Soft drinks (43.7%), crisps (43.3%), bread (42.6%), biscuits (41.6%), doughnuts/cake (37.9%), fries (37.8%) and chocolates (36.1%) were most often consumed three or more times a week while watching TV (data not shown). Fruit was consumed while watching television by 30.3% three or more times a week. With regards to gender, consuming doughnuts/cakes (*p* = 0.026, male = 42.0%, female = 35.1%) and fries (*p* = 0.043, male = 41.8%, female = 35.2%) three or more times per week while watching TV was significant. Only consuming crisps three or more times per week while watching TV resulted in significant urban–rural differences (*p* = 0.030, urban = 47.1%, rural = 40.1%).

Fat cakes are fried balls of bread dough; boerewors is a local beef sausage; samosas are triangles of fried pastry with a curry meat or vegetable filling; cordial is a sweetened syrup or powder to which water is added; sweets are candy; and pita bread is a flat round bread which is usually eaten with a filling inside.

### 3.6. Sources of Nutrition Information

According to Figure 3, most learners received nutrition information from school lessons (72%), followed by television (69.4%) and books (67.8%). Less than 40% indicated receiving information while working in the school garden, from a father or grandfather or from friends. Significant gender differences were noted for mother as source of information (*p* = 0.002), school lessons (*p* = 0.001), radio (*p* = 0.011), by television (*p* < 0.001) and books (*p* = 0.031), with percentages being significantly higher in females, while a higher percentage of males received nutrition advice from fathers/grandfathers (*p* = 0.021). Significant urban–rural differences were noted in the following: during lessons (*p* < 0.001), working in the school garden (*p* < 0.001), during the school meal (*p* = 0.020), and radio (*p* = 0.015), with percentages being higher in rural areas.

### 3.7. Variables Associated with Weight Status

Table 5 shows a significant difference in BMI categories between males and females with males having a higher prevalence in the underweight and normal categories compared to females, and females having a higher prevalence in the overweight and obese categories compared with males (*p* < 0.001).

Compared to females, males had a significantly higher risk of being underweight (OR = 2.54; *p* < 0.01, and a significantly lower risk of being either overweight or obese (OR = 0.20; *p* < 0.01) or being obese (OR = 0.18; *p* < 0.01). When combining overweight and obese learners, those in grade 12 had a significantly higher risk than other grades (OR = 1.95; *p* < 0.05). Protective factors against overweight and obesity combined and obesity on its own were males and, respectively. Compared to learners whose mothers had no formal education, learners whose mothers had grade 12 education (OR = 1.77; *p* < 0.05) or tertiary education (OR = 2.14; *p* < 0.05) had were more likely to fall in the combined overweight or obese category. There were no significant differences regarding fathers’ level of education or between the three age groups.

The only significant dietary practices associated with weight status was usual weekly consumption of breakfast, with those in the overweight category having it significantly more frequently than those in the underweight category (Appendix A). When overweight and obesity are combined, this becomes even more significant (*p* = 0.003). There were no other significant predictors of risk of underweight, overweight and obesity combined or risk of obesity.

## 4. Discussion

The present study, which was completed in disadvantaged schools in the Eastern Cape Province in South Africa, showed urban–rural and gender differences in both the eating practices and weight status of adolescents. Most notable is the high prevalence of overweight and obesity in the female adolescents (36.1%).

Gender differences in the prevalence of obesity have also been found in numerous studies done in South African adults, including in young adults and adolescents [4,5,28,29]. Significant gender differences have also been found in other sub-Saharan countries [30,31,32,33] and in African Americans [34], with females showing significantly higher levels of overweight/obesity compared with males. Case and Menendez [35] have postulated that three factors are responsible for this gender difference by studying black Africans in an urban township in Cape Town. These include the finding that women who were nutritionally deprived as children were more prone to obesity, the fact that males did not appear to have the same risk, and the finding that women of greater socio-economic status were more likely to be obese than those who were not [33,34,35].

There were also urban–rural differences with learners in urban areas of the study having a higher prevalence of overweight and obesity and rural learners having a lower prevalence in those categories. When the overweight and obesity categories were combined, urban learners in the present study had a significantly greater prevalence of overweight/obesity. This has also been shown in the recent South African Demographic and Health Survey (SADHS) [29]. Historically, black communities have faced different public health problems in urban and rural areas, with urban dwellers being more prone to non-communicable diseases, while communicable diseases are still more prevalent in rural areas [36].

A review by Micklesfield et al. [37] showed a schematic representation of the inter-relationships between the social, behavioral and environmental determinants of black South African women. They postulate that the most important factors influencing weight status of black women are gender, education, urbanization and socio-economic status. Centered within these factors are diet and eating behavior, physical activity, early life factors and body image.

The diet and eating behavior of learners was assessed in this study. Females had a significantly higher frequency of eating sugary snacks than males. They also were less likely to eat breakfast on weekdays. Overall, however, both males and females had a frequent consumption of energy-dense snacks high in added sugar, fat, salt and SSBs. Urban learners consumed more SSBs than rural ones. Soft drinks, crisps, biscuits, doughnuts/cake, fries, and chocolates were eaten by a large majority three or more times a week while viewing TV.

Learners in overweight and obesity combined categories in the present study had mothers with a significantly higher education. There were no significant differences regarding fathers’ level of education or between the three age groups. Level of education, although highly related to socio-economic status, has been shown to be independently associated with obesity in South Africa and other SSA countries. Studies in many SSA countries [32,38,39], as well as regions in South Africa with lower socio-economic status [40], have shown a positive association between level of education and obesity. Socio-economic status also influences eating behaviors and low status is associated with increased levels of obesity [12].

Costa et al. [41] reviewed 26 studies which examined the association between consumption of ultra-processed foods (defined as snacks, fast foods, junk foods and convenience foods) as well as specific ultra-processed foods such as soft drinks, sweets, chocolates, and ready to eat cereals, and body fat during childhood. Their review showed that most of the studies showed a positive association between ultra-processed foods and body fat during adolescence. Allman-Fanereli et al. [42] reported that younger generations are becoming heavier sooner than their parents. They are also the highest consumers of SSBs and fast foods and lowest consumers of vegetables and fruit among all adult groups. Larson and Story [43] examined the relationship between snacking behavior in adolescence and weight status. They found that frequent snacking is associated with a higher intake of total energy and energy from sugars (added and total). When they assessed the availability of snacks, it was noted that that energy-rich low nutrient snacks are widely available in settings/places where youth are found. 

Moreno and Rodriquez [44] found correlations between overweight in childhood and the following practices: eating supper while watching TV, consuming more energy at dinner or less at breakfast, missing breakfast, and buying lunch at school. Gorely et al. [11] found that some variables were consistently associated with TV viewing. Positive associations included being white, body weight, between meal snacking, parent’s TV habits, weekends, and having a TV in the bedroom. Some of these practices were also observed in the present study. Studies in rural areas of Kenya and Tanzania indicated that learners consumed high levels of sugar, particularly in black tea. This is likely to contribute to the development of overweight/obesity [20]. In the present study, significantly more urban learners had hot drinks and added more sugar to their drinks.

It is well known that the double burden of malnutrition is found in many low- and middle- income countries and differs by socio-economic status [19,45]. Many changes which influence weight status and encourage the development of overweight and obesity are the very rapid changes in the food systems, such as the availability of cheap ultra-processed foods and sweetened beverages, as also found in the present study. This is generally combined with large reductions in physical activity at home, transport and at work. Although physical activity was not tested in the present study, in the SANHANES [5], 43.4% of black females and 25.4% of males aged 18–24-years were found to be unfit when tested. Thus, when considering the poor dietary practices of females in the present study and the physical activity figures from the SANHANES, it is unsurprising that the prevalence of combined overweight and obesity was found to be so high in females. However, the only significant dietary practices associated with weight status in the present study was usual weekly consumption of breakfast, with those in the overweight category having breakfast significantly more frequently than those in the underweight category. This is similar to a study by Sedibe et al. [46] who studied black adolescents in urban and rural settings in South Africa. They found that the only dietary habits and eating practices within the home environment were associated with increased risk of overweight/obesity and not those in the school or community.

Since childhood and adolescent obesity has become so invasive globally, researchers agree that prevention is the best solution [2]. Primary prevention is recommended in educating children and their families about a healthy diet and enough physical activity, while secondary prevention should be aimed at lessening the effects of childhood obesity to prevent it tracking into adolescence and adulthood [2]. Keding [20] recommends that interventions need to start at school and also with adolescent and pregnant women to target the 1000-day window [47]. Furthermore, researchers need to take into consideration important gender and urban–rural differences when planning interventions since their dietary practices differ in many respects.

### Limitations of the Study

This study was only undertaken in one province, so it is not possible to generalize to other provinces. Furthermore, the shortened food frequency questionnaire did not include all food items eaten in the day. Although we refer to adolescents throughout, the sample also included some young adults. Finally, we only examined dietary practices and not sedentary behaviour, which is also a major contributor to developing obesity, among other factors [17].

## 5. Conclusions

The present study found a high prevalence of combined overweight and obesity in black female adolescents when compared with males and a high prevalence of poor dietary practices related to consumed foods and beverages. The high frequency of eating snacks in both males and females is of some concern, particularly since they mainly represent energy-dense type of foods high in fat and/or added sugar. The numerous gender and urban–rural differences in eating practices need to be taken into consideration when planning prevention and management interventions.

## Figures and Tables

**Figure 1 ijerph-17-05864-f001:**
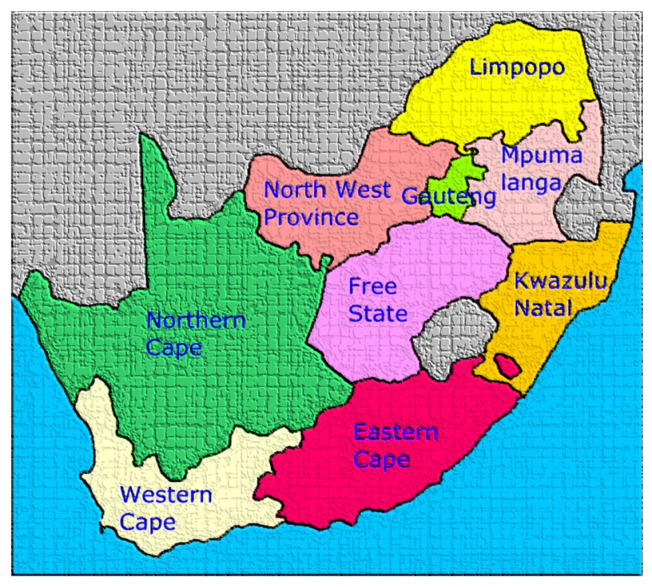
Map of South Africa showing the Eastern Cape Province.

**Figure 2 ijerph-17-05864-f002:**
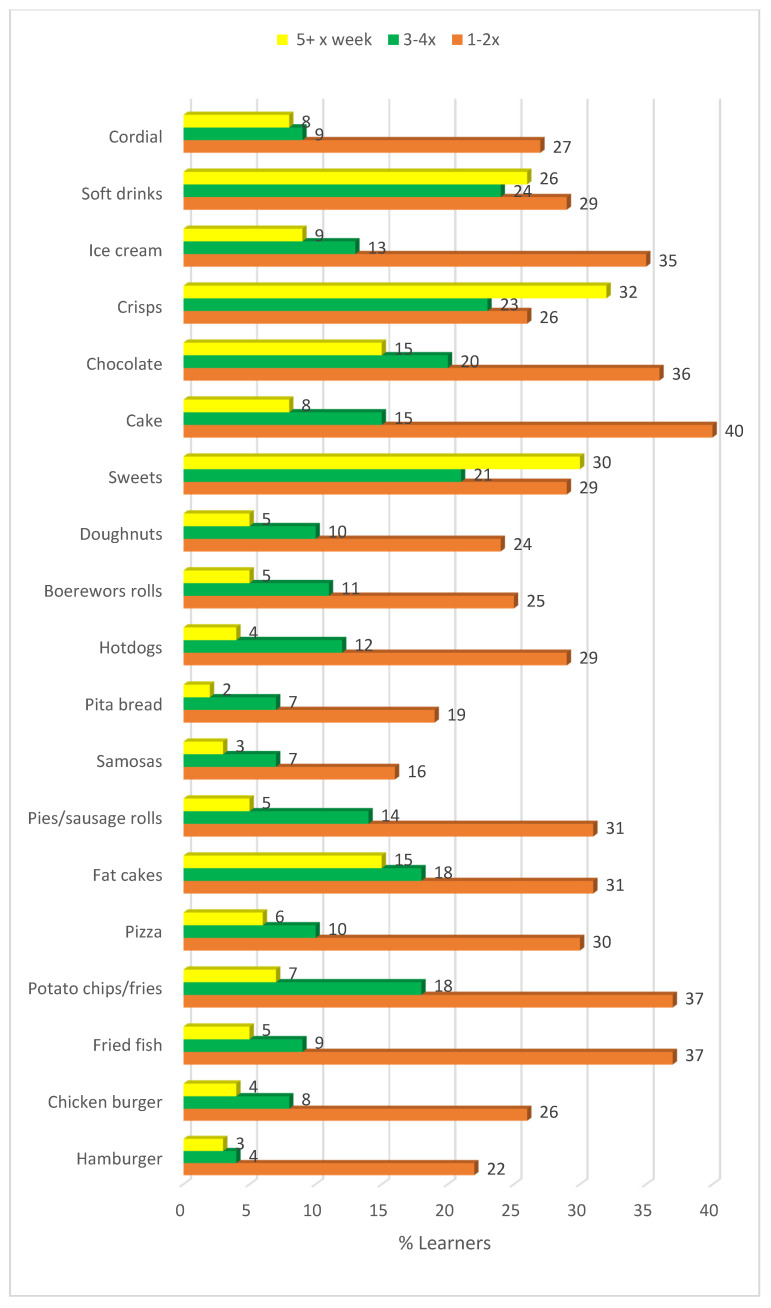
Food items reported to be purchased as take-aways during the past week by learners.

**Figure 3 ijerph-17-05864-f003:**
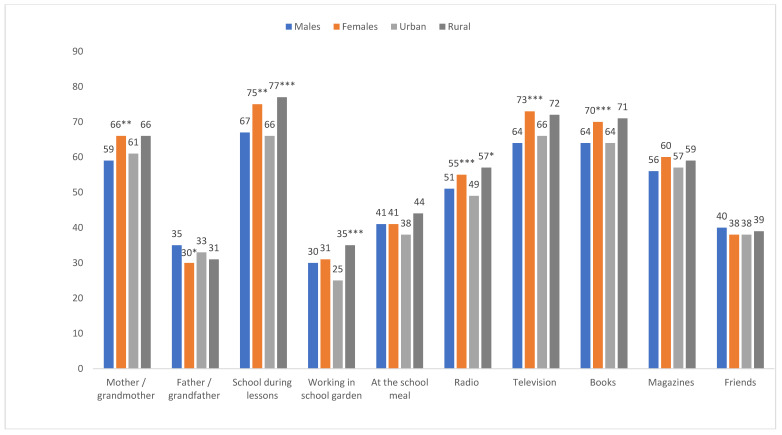
Sources of nutrition information for learners by gender and urban rural locations (figures take to nearest whole number). * Statistically significant at *p* < 0.05; ** *p* < 0.01; *** *p* < 0.001 (Rao–Scott chi-square test).

**Table 1 ijerph-17-05864-t001:** Demographic characteristics of grade 8 to 12 learners from 16 secondary schools in the Eastern Cape (N = 1360).

Variables	Variable	n	%
Age (years)	11–14	225	16.6
	15–17	606	44.7
	18–26	526	38.8
Gender	Male	529	38.9
	Female	831	61.1
Race	Black African	1315	96.8
	White	7	0.5
	Mixed ancestry	35	2.6
	Indian	2	0.1
Geographic location	Urban	653	48.0
	Rural	707	52.0
School quintiles	1 and 2	612	45.0
	3	748	55.0
Grade	8	250	18.6
	9	188	14
	10	336	24.9
	11	381	28.3
	12	192	14.3
Mother’s highest education	None	88	6.5
	Primary	220	16.2
	High school	826	60.8
	Tertiary	220	16.2
	Don’t know	5	0.4
Father’s highest education	None	150	11.1
	Primary	211	15.6
	High school	692	51.1
	Tertiary	286	21.1
	Don’t know	14	1.0

**Table 2 ijerph-17-05864-t002:** Anthropometric characteristics of secondary school learners in Eastern Cape showing significant differences between age groups and geographic location.

	Males (%)								Females (%)							
Age	All Males	11–14	15–17	18–26	*p*-Value	Urban	Rural	*p*-Value	All Females	11–14	15–17	18–26	*p*-Value	Urban	Rural	*p*-Value
n	474	78	173	223		221	253		764	127	369	268		320	444	
Mean BMI kgm^2^ (SD)	20.7 ^&&&^ 0.18	19.3 (A) 0.25	20.4 (B) 0.26	21.4 (C) 0.21	>0.05	21.0 0.24	20.4 0.22	>0.05	23.9 0.26	21.7 [B] 0.46	23.9 [A] 0.34	24.9 [A] 0.33	*p* < 0.05	24.3 0.35	23.6 0.34	>0.05
% Underweight	13.3	10.3	15.6	12.6	0.124	13.6	13.0	0.014 *	5.5	6.3	6.0	4.5	0.402	6.3	5.0	0.070
% Normal weight	76.8	75.6	76.9	77.1		72.4	80.6		58.4	66.1	58.5	54.5		52.5	62.6	
% Overweight	7.4	12.8	4.6	7.6		10.0	5.1		23.8	19.7	23.8	25.7		25.3	22.7	
% Obese	2.5	1.3	2.9	2.7		4.1	1.2		12.3	7.9	11.7	15.3		15.9	9.7	
% Overweight and obese	9.9	14.1	7.5	10.3	<0.05 #	14.0	6.3	<0.001 ***	36.1	27.6	35.5	41.0	0.053	41.3	32.4	<0.023 $

^&&&^ Significant difference between gender groups, independent *t*-test, *p* < 0.001. # Significant relationship between age and overweight/obesity for males, chi-square test, *p* < 0.05. * Significant relationship between area of residence and BMI status, males, chi-square test, *p* < 0.05. *** Significant relationship between area of residence and overweight/obesity for males, chi-square test, *p* < 0.0001. $ Significant relationship between area of residence and overweight/obesity for females, chi-square test, *p* < 0.05. (A), (B) and (C): Different symbols indicate significant differences between age groups, Bonferroni, *p* < 0.05.

**Table 3 ijerph-17-05864-t003:** The usual meal pattern and frequency of beverages and snacks consumed for learners in secondary schools in the Eastern Cape as described by a food frequency questionnaire (N = 1360).

	Frequency	Total (%)	Gender			Place of Residence		
		**n = 1336**	**Male** **n (%)**	**Female** **n (%)**	**Rao–Scott Chi-sq. *p*-Value**	**Urban n (%)**	**Rural n (%)**	**Rao–Scott Chi-sq. *p*-Value**
Number days breakfast is usually eaten on weekdays	≤2	435 (32.6)	151 (29.1)	284 (34.8)	0.002 **	241 (37.9)	194 (27.7)	<0.001 ***
	3–5	901 (67.4)	368 (70.9)	533 (65.2)		395 (62.1)	506 (72.3)	
Number of days breakfast is usually eaten on weekends	≤1	369 (27.5)	153 (29.3)	216 (26.3)	0.339	215 (33.8)	154 (21.8)	<0.001 ***
	2	973 (72.5)	369 (70.7)	604 (73.7)		422 (66.2)	551 (78.2)	
Number of meals usually eaten on weekdays	1-2	530 (40.2)	212 (41.5)	318 (39.5)	0.684	237 (37.9)	293 (42.4)	0.359
	≥3	787 (59.8)	299 (58.5)	488 (60.5)		389 (62.1)	398 (57.6)	
Number of meals usually eaten on weekends	1-2	507 (38.1)	212 (41.1)	295 (36.3)	0.189	239 (38.1)	268 (38.2)	0.992
	≥3	822 (61.9)	304 (58.9)	518 (63.7)		388 (61.9)	434 (61.8)	
Number of times snacks are usually eaten per day	≤2	944 (70.1)	397 (75.8)	547 (66.5)	0.008 **	444 (69.1)	500 (71.0)	0.601
	≥3	403 (29.9)	127 (24.2)	276 (33.5)		199 (30.9)	204 (29.0)	
Number of times of drinking hot drinks# per day	≤2	1080 (80.2)	416 (79.4)	664 (80.7)	0.701	489 (76.0)	591 (83.9)	<0.001 ***
	≥3	267 (19.8)	108 (20.6)	159 (19.3)		154 (24.0)	113 (16.1)	
Number of teaspoons added to hot drinks	0–1	321 (24.3)	117 (22.9)	204 (25.2)	0.264	121 (19.5)	200 (28.7)	<0.001 ***
	≥2	999 (75.7)	395 (77.1)	604 (74.8)		501 (80.5)	298 (71.3)	
		**Mean (CI)**	**Mean (CI)**	**Mean (CI)**	***p*-Value**	**Mean (CI)**	**Mean (CI)**	***p*-Value**
Frequency of eating breakfast on weekdays		3.1 (3.0–3.2)	3.2 (3.1–3.3)	3.0 (3.0–3.1)	0.005 **	3.0 (2.8–3.1)	3.2 (3.1–3.3)	<0.001 ***
Frequency of taking a lunch box to school in past week		2.1 (1.9–2.3)	1.7(1.6–1.9)	2.3 (2.1–2.5)	<0.001 ***	2.2 (2.0–2.3)	2.0 (1.7–2.3)	0.358
Frequency of takeaways (past week)		3.9 (3.6–4.2)	3.7 (3.4–4.1)	4.0 (3.7–4.4)	0.166	4.0 (3.5–4.5)	3.9 (3.4–4.3)	0.679
Fried and fatty snacks (past week)		3.1 (2.9–3.2)	2.9 (2.7–3.1)	3.2 (2.9–3.4)	0.108	3.1 (2.9–3.3)	3.1 (2.7–3.4)	0.794
Sugary snacks (past week)		7.2 (6.7–7.6)	6.5(6.1–6.9)	7.6 (6.9–8.3)	0.007 **	6.9 (6.2–7.7)	7.4 (6.7–8.2)	0.274
Salty snacks (past week)		1.6 (1.5–1.8)	1.5 (1.3–1.7)	1.7 (1.6–1.9)	0.014 *	1.6 (1.3–1.9)	1.7 (1.5–1.9)	0.631
Healthy TV snacks (past week)		2.4 (1.9–2.8)	2.5 (2.0–3.1)	2.2 (1.9–2.6)	0.045 *	2.4 (1.4–3.4)	2.3 (2.0–2.7)	0.854
Unhealthy TV snacks (past week)		8.5 (7.1–10.0)	9.1 (7.3–10.8)	8.1 (6.9–9.4)	0.019 *	8.3(4.9–11.8)	8.7 (7.4–9.9)	0.809

* Statistically significant at *p* < 0.05; ** statistically significant at *p* < 0.01; *** statistically significant at *p* < 0.001; # hot drinks usually tea or coffee. Takeaways include hamburgers, chicken burgers, pizzas, pies/sausage rolls, pita bread, hot dog rolls, boerewors (beef sausage) roll; fried foods include fried fish, fries, fat cakes (fried balls of dough), samosas; sugary snacks include sweets, cake, chocolate, ice cream, doughnuts, soft drinks; salty snacks include crisps; healthy TV snacks include fruit and bread; unhealthy TV snacks include popcorn, chocolates, crisps, biscuits, cakes/doughnuts/eclairs, cold drinks, fries.

**Table 4 ijerph-17-05864-t004:** Usual breakfast consumption patterns during school days for learners in secondary schools in the Eastern Cape (N = 1360).

	Frequency	Total n (%)	Gender			Place of Residence		
			Male n (%)	Female n (%)	Rao–Scott Chi-sq. *p*-Value	Urban n (%)	Rural n (%)	Rao–Scott Chi-Square *p*-Value
Frequency of eating cooked porridge with milk for breakfast	0–2	1001 (74.8)	385 (73.9)	616 (75.3)	0.643	467 (73.7)	534 (75.7)	0.294
	3–5	338 (25.2)	136 (26.1)	202 (24.7)		167 (26.3)	171 (24.3)	
Frequency of eating cooked porridge without milk for breakfast	0–2	1107(83.7)	415 (80.3)	692 (85.9)	*p* < 0.001 ***	521 (84.3)	586 (83.1)	0.564
	3–5	216 (16.3)	102 (19.7)	114 (14.1)		97 (15.7)	119 (16.9)	
Frequency of eating instant cereal with milk for breakfast	0–2	941 (71.0)	382 (73.7)	559 (69.3)	0.269	431 (69.4)	510 (72.4)	0.237
	3–5	384 (29.0)	136 (26.3)	248 (30.7)		190 (30.6)	194 (27.6)	
Frequency of eating white bread with spread for breakfast	0–2	942 (70.9)	380 (73.2)	562 (69.4)	0.098	442 (70.8)	500 (70.9)	0.976
	3–5	387 (29.1)	139 (26.8)	248 (30.6)		182 (29.2)	205 (29.1)	
Frequency of eating white bread without spread for breakfast	0–2	1100 (83.1)	425 (82.2)	675 (83.7)	0.593	522 (84.5)	578 (82.0)	0.144
	3–5	223 (16.9)	92 (17.8)	131 (16.3)		96 (15.5)	127 (18.0)	
Frequency of eating brown bread with spread for breakfast	0–2	893 (67.4)	350 (67.2)	543 (67.5)	0.880	410 (66.1)	483 (68.5)	0.386
	3–5	432 (32.6)	171 (32.8)	261 (32.5)		210 (33.9)	222 (31.5)	
Frequency of eating brown bread without spread for breakfast	0–2	1075 (81.3)	31.6 (81.0)	657 (81.4)	0.859	510 (82.5)	565 (80.1)	0.106
	3–5	248 (18.7)	98 (19.0)	150 (18.6)		108 (17.5)	140 (19.9)	
Frequency of drinking milk or yoghurt for breakfast	0–2	914 (68.8)	364(70.4)	550 (67.7)	0.110	415 (66.5)	499 (70.8)	0.066
	3–5	415 (31.2)	153 (29.6)	262 (32.3)		209 (33.5)	206 (29.2)	
Frequency of drinking tea or coffee with milk for breakfast	0–2	822 (61.7)	310 (59.5)	512 (63.1)	0.090	413 (65.9)	409 (58.0)	0.003 **
	3–5	510 (38.3)	211 (40.5)	299 (36.9)		214 (34.1)	296 (42.0)	
Frequency of drinking tea or coffee without milk for breakfast	0–2	996 (75.4)	391 (75.8)	605 (75.2)	0.794	472 (76.6)	524 (74.3)	0.320
	3–5	325 (24.6)	125 (24.2)	200 (24.8)		144 (23.4)	181 (25.7)	
Frequency of eating fruit for breakfast	0–2	712 (53.4)	284 (54.7)	428 (52.6)	0.516	346 (55.1)	366 (51.9)	0.541
	3–5	621 (46.6)	235 (45.3)	386 (47.4)		282 (44.9)	339 (48.1)	

** *p* significant at *p* < 0.01; *** *p* significant at *p* < 0.001; n = number of individual characteristics.

**Table 5 ijerph-17-05864-t005:** Socio-demographic factors associated with nutritional status using BMI as an indicator and bivariate logistic regression with BMI status as outcome.

Factors	BMI Status					Bivariate Logistic Regression		
	Underweight n (%)	Normal weight n (%)	Overweight n (%)	Obese n (%)	Rao–Scott Chi-Square *p*-Value	Risk of Underweight OR (95% CI)	Risk of Overweight or/Obesity OR (95% CI)	Risk of Obesity OR (95% CI)
**Grade**						n = 105 UW	n = 323 OV + O	n = 106 O
8	17 (7.6)	165 (74.0)	31 (13.9)	10 (4.5)	0.1681	Ref	Ref	Ref
9	17 (10.0)	110 (64.7)	29 (17.1)	14 (8.2)		1.28 (0.47–3.50)	1.33 (0.71–2.48)	1.68 (0.63–4.49)
10	34 (11.1)	193 (63.3)	47 (15.4)	31 (10.2)		1.45 (0.62–3.40)	1.35 (0.87–2.11)	2.13 (1.04–4.33) *
11	22 (6.4)	228 (65.9)	66 (19.1)	30 (8.7)		0.79 (0.29–2.14)	1.51 (0.92–2.47)	1.79 (0.70–4.59)
12	13 (7.2)	110 (60.8)	39 (21.5)	19 (10.5)		0.93 (0.46–1.91)	1.94 (1.05–3.58) *	2.30 (0.81–6.54)
**Gender**								
Male	63 (13.3)	364 (76.8)	35 (7.4)	12 (2.5)	<0.001 ***	2.54 (1.47–4.39) **	0.20 (0.13–0.31) ***	0.18 (0.08–0.44) **
Female	42 (5.5)	446 (58.4)	182 (23.8)	94 (12.3)		Ref	Ref	Ref
**Mother’s highest level education**								
None	8 (10.1)	60 (75.9)	8 (10.1)	3 (3.8)	0.655	Ref	Ref	Ref
Primary	15 (7.4)	135 (66.2)	38 (18.6)	16 (7.8)		0.79 (0.27–2.27)	1.86 (0.92–3.76)	1.77 (0.45–6.95)
High school	66 (8.8)	488 (65.2)	132 (17.6)	63 (8.4)		0.93 (0.45–1.94)	1.77 (1.10–2.83) *	1.86 (0.49–7.11)
Tertiary	16 (8.0)	125 (62.2)	37 (18.4)	23 (11.4)		0.84 (0.37–1.92)	2.14 (1.09–4.23) *	2.63 (0.56–12.41)
**Father’s highest level of education**								
None	15 (10.6)	93 (66.0)	24 (17.0)	9 (6.4)	0.992	Ref	Ref	Ref
Primary	14 (6.9)	135 (66.8)	35 (17.3)	18 (8.9)		0.60 (0.28–1.28)	1.17 (0.62–2.21)	1.50 (0.55–4.12)
High school	51 (8.3)	401 (65.3)	109 (17.8)	53 (8.6)		0.68 (0.34–1.35)	1.07 (0.73–1.57)	1.34 (0.46–3.90)
Tertiary	22 (8.4)	169 (64.8)	45 (17.2)	25 (9.6)		0.71 (0.39–1.30)	1.13 (0.64–2.00)	1.54 (0.40–5.97)
**School location**								
Urban	50 (9.2)	328 (60.6)	103 (19.0)	60 (11.1)	0.020 *	Ref	Ref	Ref
Rural	55 (7.9)	482 (69.2)	114 (16.4)	46 (6.6)		1.02 (0.59–1.76)	0.88 (0.64–1.21)	0.69 (0.36–1.30)

OR = Odds ratio. * Statistically significant at *p* < 0.05; ** *p* < 0.01; *** *p* < 0.001; O = obese, OV = overweight; UW = underweight.

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
