# Peer review of "Dietary Practices and Adolescent Obesity in Secondary School Learners at Disadvantaged Schools in South Africa: Urban–Rural and Gender Differences"

_ijerph, 2020, doi:10.3390/ijerph17165864_

Round 1

Reviewer 1 Report

The authors present the article entitled "Dietary Practices and Adolescent Obesity in Secondary School Learners at Disadvantaged Schools in South Africa: Urban-rural and gender differences". It seems an interesting article mainly by the wide sample. However, there specific aspects, mainly in the methodology that must be solved.

Introduction

The authors present an introduction with several paragraph across to this section that offer repetitive concepts.

The following phrase should be included in the methods section and not after the aim: "Dietary practices are defined as: “observable actions or behaviors of dietary habit and can be classified as good dietary practices and poor dietary practices [21.”

Methods

Why the authors select 16 schools?

Why the number of female was higher than the males?

Authors explain that they used a short questionnaire based on "Keiron A. Audain, Frederick J. Veldman, Susanna M. Kassier" study. However, this study do not seem the validation of a questionnaire. In addition, they express that "The items selected were based on foods typically eaten for breakfast in the Eastern Cape". It could be a limitation of the study due to if you restrict the multiple options before to star the evaluation, you may are limiting the options of answer and in turn, it is missing important information.

It seems that the options of answer are focused mainly to hypercaloric nurishment. However, there are more food which can increase the caloric intake and therefore, increase the average daily intake.

Authors talk about adolescent but they did not studied the sexual maturity and stablised a range of age from 11 to 26. It is very possible that there was chindren and young-adult included in the sample, so it is not posible to talk about adolescents.

Results

Figure 2. The numbers included in the body of the figure must be delete to do the figure more visual.

The table 4 no show title.

Table 5. not all variable are well explain.

Discussion

The redaction of the discussion must be improve.

Author Response

Reviewer 1

Many thanks for your valuable input. We have tried to address all your comments

"Dietary practices are defined as: “observable actions or behaviors of dietary habit and can be classified as good dietary practices and poor dietary practices [21].” Should be moved to methods

Line 142-143: Moved to methods section.

Dietary practices are defined as: “observable actions or behaviors of dietary habit and can be classified as good dietary practices and poor dietary practices [22].”

Methods

Why the authors select 16 schools?

Why the number of female was higher than the males?

Authors explain that they used a short questionnaire based on "Keiron A. Audain, Frederick J. Veldman, Susanna M. Kassier" study. However, this study do not seem the validation of a questionnaire. In addition, they express that "The items selected were based on foods typically eaten for breakfast in the Eastern Cape". It could be a limitation of the study due to if you restrict the multiple options before to start the evaluation, you may are limiting the options of answer and in turn, it is missing important information.

It seems that the options of answer are focused mainly to hypercaloric nurishment. However, there are more food which can increase the caloric intake and therefore, increase the average daily intake.

Authors talk about adolescent but they did not studied the sexual maturity and stablised a range of age from 11 to 26. It is very possible that there was chindren and young-adult included in the sample, so it is not posible to talk about adolescents.

Results

Figure 2. The numbers included in the body of the figure must be delete to do the figure more visual.

The table 4 no show title.

Table 5. not all variable are well explain.

Methods

Line  108-115 : regarding number of schools:

The third stage involved randomly selecting 18 eligible secondary schools from quintiles 1, 2 and 3 in each education district. It was calculated that this number of schools would provide a representative sample.  The fourth stage involved purposive selection of grade level 8 to 12 in each of the selected schools. These grade levels were selected because the target population was secondary school learners. In each of the selected grades, simple random balloting was used to select learner participants.  Learners were asked to pick a “Yes” and a “No” paper in randomized manner. Those who had picked “Yes” were given a consent form to be signed by a parent/guardian. Two out of the 18 sample schools were eliminated from the study because learners did not return the consent forms.

Regarding more females than males see line 118-120: More male learners than female learners did not return their consent forms and were excluded from the study; resulting in an unequal distribution. 

Regarding the Audain questionnaire

Line 147-150: We added: The original questionnaire had been developed by six dietitians and provided options for 61 foods according to the following groups: starches, vegetables, fruit, dairy, meat, fast food, takeaways, snacks and drinks. Foods included in the current FFQ were those eaten for breakfast, in the school environment, takeaways, fast foods, snacks and drinks. However, the reviewer has made an important point and this has been added to the limitations:

Line 495-497 we added:

This study was only undertaken in one province, so it is not possible to generalize to other provinces. Furthermore, the shortened food frequency questionnaire did not include all food items eaten in the day.

 Regarding hypercaloric nourishment:

We were interested in looking at unhealthy practices of adolescents particularly those which could be associated with obesity which is why we focused on energy-dense foods, particularly those high in fat and sugar

Regarding age of learners

Line 220-224: The World Health Organization (WHO) defines adolescents as those people between 10 and 19 years of age [27]. “Other overlapping terms used in this report are youth (defined by the United Nations as 15–24 years) and young people (10–24 years), a term used by WHO and others to combine adolescents and youth [27].” For the purpose of this study we will refer to adolescents since they made up the major part of the sample.

Line 494-496: Although we refer to adolescents throughout the article the sample also comprised some young adults.

Results

Figure 2 has been adapted to show numbers more clearly

Table 4. We have removed the contents of Table 4 and added it to Table 3

Table 5 (Now 6) we have removed two columns and revised it to improve clarity

Reviewer 2 Report

In this manuscript, the authors studied the dietary practices and weight status of male and female adolescents in the Eastern Cape province in urban and rural areas. They showed higher prevalence of combined overweight and obesity, and increased prevalence of poor eating practices in black female adolescents. They also showed significant differences in eating practices in urban-rural areas. Overall, their findings are interesting, and may be of clinical use.

There are several minor concerns regarding the manuscript:

-In the samples, there are 38.9% males and 61.1% females. What is the distribution of male and female learners among the ages? How representative are the samples for this type of study, since there is a large percentage difference between male and female learners?

-What is correlation between BMI and eating practices?

-What are the effects of other possible factors that may contribute to the different weight status and eating practices, such as exercise time, sleeping hours or time between food consumptions?

Author Response

Reviewer 2

Your comments are much appreciated, and we have done our best to address them

In this manuscript, the authors studied the dietary practices and weight status of male and female adolescents in the Eastern Cape province in urban and rural areas. They showed higher prevalence of combined overweight and obesity, and increased prevalence of poor eating practices in black female adolescents. They also showed significant differences in eating practices in urban-rural areas. Overall, their findings are interesting, and may be of clinical use.

There are several minor concerns regarding the manuscript:

-In the samples, there are 38.9% males and 61.1% females. What is the distribution of male and female learners among the ages? How representative are the samples for this type of study, since there is a large percentage difference between male and female learners?

-What is correlation between BMI and eating practices?

-What are the effects of other possible factors that may contribute to the different weight status and eating practices, such as exercise time, sleeping hours or time between food consumptions?

 Regarding more females than males see line 118-120: More male learners than female learners did not return their consent forms and were excluded from the study; resulting in an unequal distribution. 

However, the large difference in obesity between males and females has been described in numerous other studies including those highlighted in the text.

See line 47-55:

The combined prevalence of overweight and obesity of South African adolescents increased from 29.3% to 36.5% in females and from 7.9% to 14.3% in males between 2002 and 2008, respectively [4]. In both years, the overall prevalence was substantially higher in females than males. In the South African Health and Nutrition Examination Study (SANHANES ) of 2012 [5] the combined prevalence of overweight and obesity was 26.8% in 15- to 17-year -old females and 48.8% in 18-to 24 year-old females, implying an even greater increase. A recent review of cross-sectional studies undertaken on South African adolescents in urban and rural settings showed an overall increase in the combined prevalence of overweight and obesity with age in adolescents from age 10 - to 20- years, mostly because of an increase in obesity in females.

Regarding association of BMI with eating practices

-In order to show the effects of eating practices on BMI we included a bivariate logistic regression which is Supplementary Table 1.

Line: 200-202. Bivariate logistic regression analysis was conducted to establish relationships between the risk of underweight, overweight/obesity or obesity between sociodemographic characteristics and eating practices, respectively.

Regarding other possible factors

Line 498-499 we added this to limitations

- Finally, we only examined dietary practices and not sedentary behaviour, which is also a major contributor to developing obesity, among other factors [17].

Reviewer 3 Report

Paper by Doctor Okeyo AP et al treated dietary practice and obesity in South African adolescents in schools for disadvantaged people. I would like to provide comments to improve the study manuscript.

[Major]

  1. BMI: In almost all tables, categories of BMI or means and SDs of BMI need to be indicated. The title indicates obesity. As a reader, I wonder how BMI was reflected in the analyses.
  2. I would like to request means and SDs of BMI in Table 1. Additionally, I wonder proportions of underweight, overweight and obesity.
  3. Table 2 may include too much figures and words. As a reader, I would like to be explained in simpler illustration. Specially, Words in the first column need to be diminished.
  4. Table 2 and 3 need to include BMI categories, or means and SDs of BMI.
  5. Figure 2 would need to revised for publication. Universal design is needed for international readers. As a reader, I could not understand the meanings, even with title, legends and captions.
  6. As a reader, I would like to combine the former tables and figure 4, indicating dietary characteristics of adolescents with obesity vs. underweight.

[Minor]

  1. In the Introduction section, expression of ‘male vs. female‘ and ‘urban vs. rural‘ may be informative.
  2. Table 4 may include too many words in the first column.

Overall, I consider that tables and figures of the manuscript need to be greatly revised for publication. Analyses for illustrating the solution of their study question also need to be revised. Statistical analyses would not be hurdle in this study. With modifications, the study would add evidence to epidemiology of child obesity and underweight.

Author Response

Reviewer 3

Thank you for valuable comments. We have done our best to address your comments and believe that the article is much improved

  1. BMI: In almost all tables, categories of BMI or means and SDs of BMI need to be indicated. The title indicates obesity. As a reader, I wonder how BMI was reflected in the analyses.
  2. I would like to request means and SDs of BMI in Table 1. Additionally, I wonder proportions of underweight, overweight and obesity.
  3. Table 2 may include too much figures and words. As a reader, I would like to be explained in simpler illustration. Specially, Words in the first column need to be diminished.
  4. Table 2 and 3 need to include BMI categories, or means and SDs of BMI.
  5. Figure 2 would need to revised for publication. Universal design is needed for international readers. As a reader, I could not understand the meanings, even with title, legends and captions.
  6. As a reader, I would like to combine the former tables and figure 4, indicating dietary characteristics of adolescents with obesity vs. underweight.

We have included an additional Table 2. This provides means (SDs) as well as the proportions of BMI. It also includes significant gender and urban-rural differences.

We have reduced Tables 2 and 3 (now 3 and 4) considerably by combining some categories which makes it shorter and easier to read.

In order to show the effects of eating practices on BMI we included a bivariate logistic regression which is Supplementary Table 1. This has been described in the data analyses section:

Line: 200-202. Bivariate logistic regression analysis was conducted to establish relationships between the risk of underweight, overweight/obesity or obesity between sociodemographic characteristics and eating practices, respectively.

Figure 2 has been revised by adding explanations of all local foods. The rest are all American terms: The following has been added:

Lines: 358-362

Fat cakes are fried balls of bread dough; boerewors is a local beef sausage; samosas are triangles of fried pastry with a curry meat or vegetable filling; cordial is a sweetened syrup or powder to which water is added. Soft drinks are sugar sweetened beverages

  1. In the Introduction section, expression of ‘male vs. female‘ and ‘urban vs. rural‘ may be informative.
  2. Table 4 may include too many words in the first column

We have highlighted the male and female and

urban and rural areas in the introduction

We have changed the format to improve reading of Table 4.

Round 2

Reviewer 1 Report

The authors have taken in account my recomendations, so the manuscript is more clear now and also it has been describen the possible limiitations of this.

However, I recomende to delate "genders differeces" from the title due to the sample include mainly female.

Author Response

According to our statistician, although  the sample includes fewer males (n=529) than females (n=831), the sample is large enough to detect significant differences in gender taking into account the tests we used.

It would be a great pity to remove gender from the title since there were important gender differences found in the eating patterns. For example females had a higher frequency of eating sugary snacks (p<0.001) and lower frequency of eating breakfast (p<0.01) than males. Females ate significantly more fried fish (p<0.05), pizza (p<0.05) fat cakes (fried dough balls)(p<0.05), hot dogs (p<0.01), candy (p<0.001), cake (p<0.01), and crisps (p<0.001).

Reviewer 3 Report

I consider that the researchers have addressed all of my concerns. I have  no more concern. I appreciate their efforts to report the important results. 

Author Response

The reviewers did not request any further changes.